# Shade-Induced Leaf Senescence in Plants

**DOI:** 10.3390/plants12071550

**Published:** 2023-04-04

**Authors:** Zhuang Li, Tao Zhao, Jun Liu, Hongyu Li, Bin Liu

**Affiliations:** The National Key Facility for Crop Gene Resources and Genetic Improvement (NFCRI), Institute of Crop Science, Chinese Academy of Agricultural Sciences, Beijing 100081, China

**Keywords:** shade, light, photoreceptor, phytohormone, leaf senescence

## Abstract

Leaf senescence is a vital developmental process that involves the orderly breakdown of macromolecules to transfer nutrients from mature leaves to emerging and reproductive organs. This process is essential for a plant’s overall fitness. Multiple internal and external factors, such as leaf age, plant hormones, stresses, and light environment, regulate the onset and progression of leaf senescence. When plants grow close to each other or are shaded, it results in significant alterations in light quantity and quality, such as a decrease in photosynthetically active radiation (PAR), a drop in red/far-red light ratios, and a reduction in blue light fluence rate, which triggers premature leaf senescence. Recently, studies have identified various components involved in light, phytohormone, and other signaling pathways that regulate the leaf senescence process in response to shade. This review summarizes the current knowledge on the molecular mechanisms that control leaf senescence induced by shade.

## 1. Introduction

Light is a vital factor in the growth, morphology, and development of plants [1]. Through photosynthesis, plants convert light energy into chemical energy, thus forming the foundation of all life on our planet [2]. As the Earth rotates and revolves, the light environment (light intensity and quality) changes in different seasons and at different times of the day [3]. Since plants are sessile organisms incapable of long-distance migration, they have developed the ability to detect changes in light intensity and quality and continuously adjust their growth and development to ensure their survival in variable light environments [4]. When shaded by other plants or grown under high-density planting conditions, plants undergo adaptive architectural changes to capture light through phototropism and shade avoidance syndrome (SAS), such as increased stem and petiole elongation, hyponastic leaf orientation, reduced branching, increased apical dominance, and accelerated leaf senescence [5,6,7,8].

Plants possess an intricate array of photoreceptors that respond to different wavelengths of light, ranging from near-ultraviolet B (UVB) to far-red (FR) light [9,10]. The phytochromes are primarily sensitive to red light (R) and a portion of the FR light spectrum (600–750 nm). The phytochrome-mediated signaling process is based on the transformation between an inactive Pr form, which absorbs red light, and an active Pfr form, which absorbs FR light [11,12]. Additionally, plants possess three types of flavin-based blue light receptors: cryptochromes, phototropins, and three LOV/F-box/Kelch-repeat proteins: Zeitlupe (ZTL), flavin-binding Kelch repeat F-BOX1 (FKF1), and LOV Kelch repeat protein 2 (LKP2) [13,14,15]. Exposure to blue light induces conformational changes in cryptochromes, such as the separation of the PHR and CCE domains, which leads to the formation of cryptochrome homo-oligomers that interact with cryptochrome-interacting proteins to regulate blue light-mediated gene expression and plant growth [16]. UVR8 is responsible for the absorption of UVB light [17]. In the darkness, UVR8 is present as a homodimer; however, when exposed to UVB light, it breaks apart into monomers, which then alters the expression of downstream targets [18].

Leaf senescence is the final stage of leaf development and is not a merely passive process directed by an intrinsically regulated genetic program [19]. During senescence, leaf cells experience a highly ordered alteration in gene expression, biochemical metabolism, and cell structure [20]. The earliest and most evident change in cell structure is the breakdown of the chloroplast, followed by the degradation of macromolecules such as membrane lipids, proteins, and nucleic acids [21]. This enables the recycling of exportable nutrients from the senescent leaf to other growing organs and developing seeds, thus increasing reproductive success [22]. Thus, although it has a cost to the mature leaf organ, leaf senescence can be seen as an altruistic process to guarantee optimal production of offspring and better survival of plants in given temporal and spatial conditions [19,23,24].

Although leaf senescence is mainly driven by developmental age, it is also affected by a range of internal factors, such as phytohormones, as well as external signals, including biotic and abiotic stress and light environment [19]. As such, leaf senescence is an integrated response to leaf developmental age information, as well as other internal and external stimuli. Light is one of the most influential factors in initiating leaf senescence. Especially in densely populated crop communities, light becomes a limiting factor. Recent studies have revealed the molecular mechanisms of photosynthesis-, photoreceptor-, and phytohormone-mediated leaf senescence under shade conditions. In this review, we discuss the recent advances in the molecular and genetic understanding of shade-induced leaf senescence.

## 2. Photoreceptor-Mediated Leaf Senescence under Shade Conditions

The light environment drastically changes in both quantity and quality under shade conditions [5,8]. Reduced light intensities and alterations in light composition strongly contribute to the onset of senescence for shaded plants [25,26]. When a vegetative canopy or high-density planting is encountered, the light environment experiences a reduction in the total amount of photosynthetically active radiation (PAR), as well as modifications in its spectral composition [27]. Most of the visible light (especially red and blue light) is absorbed by upper leaves, while far-red light and green light are poorly absorbed and mostly transmitted and reflected by leaves (Figure 1). Studies have indicated that the ratio of red to far-red light (R:FR) is approximately 1.15 and remains relatively stable with weather and season. Nevertheless, beneath a vegetative canopy, R:FR drops to as low as 0.09 [28].

### 2.1. Red and Far-Red Light Receptor Phytochrome

Phytochrome is a photoreceptor that perceives both red and far-red light and controls a variety of physiological processes, such as seed germination, seedling de-etiolation, shade avoidance response, and floral transition [11,29]. In the model plant Arabidopsis, there are five phytochrome genes (phyA–phyE), with phyA and phyB being the most widely studied [11]. The mechanism of phytochrome (PHYs) regulating shade-induced leaf senescence has been well-documented in Arabidopsis. Upon R illumination, the inactive Pr state photoconverts to the active Pfr state, which is then inactivated upon FR light absorption or through a process of temperature-dependent thermal relaxation (Figure 1). In environments with high planting density or under the vegetational canopy, where the R/FR ratio drops substantially, this leads to reduced phytochrome activity [29].

The increased inactive Pr form of phytochrome leads to the dephosphorylation of phytochrome interacting factors (PIFs), which are basic helix-loop-helix (bHLH) transcription factors (TFs) that act in the signaling pathway of leaf senescence, thus increasing their protein stability [11,30]. The *pif* quadruple mutant (*pif1, pif3, pif4, and pif5*) *pifQ* exhibited slower senescence than wild-type plants under attached, detached, and individually covered leaves conditions, indicating that PIF proteins are positive regulators of leaf senescence under shade conditions [30,31,32,33]. Numerous downstream target genes of PIFs implicated in the leaf senescence pathway have been identified. PIF4 and PIF5 have been found to bind directly to the promoter of *ORE1*, a NAC (NAM, ATAF, and CUC) transcription factor that positively regulates senescence in Arabidopsis leaves, and activates its transcription [30,34]. Furthermore, PIF4 and PIF5 have been shown to directly activate the transcription of two senescence-promoting phytohormones (ethylene and ABA) signaling components [30,35]. Interestingly, the *pif5* mutant exhibited a significant increase in amino acid accumulation in the darkened leaves compared to the wild-type plants. The metabolic shift in the amino acid pool prolongs the lifespan of leaves by facilitating respiratory substrates and enhancing polyamines biosynthesis. This, in turn, retards leaf senescence by interfering with the ethylene signaling pathway [36].

Chloroplasts are the energy factories in plant leaves and significantly influence photosynthesis, leaf development, and senescence. PIF4 and PIF5 have been shown to directly activate the transcription of yellow coloring *1* (*NYC1*), subunits of chlorophyll b reductase that catalyze the conversion of chlorophyll b to chlorophyll a, and stay green (*SGR*), an interacting-partner of a subset of the proteins in the light-harvesting complex II (LHCII) [32,33]. Golden2-like 1 (GLK1) and GLK2, GARP-type MYC TFs that directly regulate genes involved in photosystem proteins and chlorophyll biosynthesis enzymes, are identified as the direct downstream targets of PIF TFs during leaf senescence [33,37]. PIF TFs were also found to directly activate senescence associated gene *29* (*SAG29*) by binding to the G-box motif in the promoter [31]. Thus, shade-enhanced PIFs proteins initiate the process of leaf senescence by promoting the chlorophyll degradation and transcription of senescence-promoting phytohormones components.

In Arabidopsis, phyA and phyB are the primary photoreactors that modulate low R:FR-induced leaf senescence in response to shade. Intermittent pulses of red light in the darkness (phyB on) delay cotyledons senescence (Figure 1). However, pulses of red light followed by far-red light in the darkness (phyB off) accelerate cotyledons senescence, indicating that the Pfr form of phyB inhibits senescence [30]. Phenotypic analysis shows that *phyB* mutants senesced faster and *PHYB-OX* plants senesced slower than wild-type plants under detachment and dark incubation conditions in Arabidopsis. Rice *phyB* T-DNA insertion knockout mutant (*OsphyB*) exhibits similar premature leaf senescence to Arabidopsis phyB mutant [38], demonstrating functional conservatism in regulating senescence among different species. In contrast to *phyB*, phyA mutant, and *PHYA-OX* plants show no delayed or accelerated senescence compared with wild-type plants under white light conditions. However, *phyA* mutant leaves turn yellowing faster than those of wild-type plants under far-red light-enriched conditions [39,40], indicating phyA regulates leaf senescence only under specific conditions, such as shade. PhyA is involved in maintaining the chlorophyll content in response to vegetational shading, and the accelerated leaf yellowing in *phyA* mutant plants is not accompanied by the increased transcription of senescence-specific molecular markers, suggesting an indirect role of phyA in inducing leaf senescence in response to vegetational canopy [39]. Thus, phyA and phyB play specific and partially overlapping functions in modulating the transcription of senescence-promoting genes and fine-tuning the chlorophyll biosynthetic pathway to initiate the process of leaf senescence in response to vegetational shading.

### 2.2. Blue Light Receptor Cryptochrome

Blue light is drastically reduced when plants are shaded by neighboring plants [5,8]. Unlike the well-documented low R:FR-induced leaf senescence, current knowledge involved in the molecular mechanism of low blue light-induced leaf senescence is rather poor. In contrast to phytochromes, cryptochromes seem to play a comparatively weak function in regulating leaf senescence in model plant Arabidopsis. The detached leaves of *cry1*, *cry2,* and *cry1cry2* double mutant did not show a delayed or premature leaf senescence during dark incubation in Arabidopsis [30]. The most recent study demonstrates that CRY2 is primarily responsible for the suppression of leaf senescence, and the function of CRY1 in inhibiting leaf senescence is comparatively weak compared with CRY2 under low blue light conditions [41]. Consistent with the function of CRY2 in Arabidopsis, GmCRY2a was identified as a negative regulator of leaf senescence in soybean. GmCRY2a interacts with GmCIB1 in a blue light-dependent manner to decrease the *WRKY53b* promoter-binding affinity of GmCIB1 and inhibit leaf senescence [42]. However, our previous study demonstrated that GmCRY1s predominantly mediate low blue light-induced stem elongation [43]. These results suggest that there is a complex regulatory mechanism of cryptochrome-mediated leaf senescence in response to shade, and the inactivation of CRY1 may be the main reason for premature leaf senescence under low blue light conditions.

PIF4 and PIF5 also act downstream of CRY1 and CRY2 and are necessary for low blue light-induced hypocotyl elongation. CRY1 and CRY2 directly interact with PIF4 and PIF5, and chromatin immunoprecipitation sequencing (ChIP-seq) results show that PIF4/5 and CRY2 occupy overlapping promoter regions, indicating that CRYs may regulate leaf senescence by modulating PIF4 and PIF5 activities [44]. CRY1 physically interacts with PIF4 in a blue light-specific manner to repress its transcription activity and regulate high temperature-mediated hypocotyl elongation in Arabidopsis. Thus, it is possible that cryptochrome also affects the initiation of leaf senescence under specific environmental stresses, such as high temperature. PIFs, the key leaf senescence enhancers, act downstream of phytochrome and cryptochrome to modulate the initiation of leaf senescence in response to shade conditions.

DELLA protein is another interacting partner of cryptochrome in a blue light-dependent manner in Arabidopsis [45,46]. DELLA proteins were shown as the negative regulators of leaf senescence and involved in light-mediated leaf senescence [47]. DELLA protein RGA-LIKE (RGL1) physically interacts with WRKY45, a leaf senescence enhancer in Arabidopsis, and repressed the transcription activation function of WRKY45 on several SAGs such as *SAG12*, *SAG13, SAG113*, and *SEN4* [48]. Low R:FR and low blue light intensity induce the degradation of nuclear, growth-inhibiting DELLA proteins [49,50]. CRY1 directly interacts with DELLA proteins and represses the GA-induced degradation of DELLA proteins under blue light illumination in Arabidopsis and wheat [45,46,51], indicating the conserved mechanism of cryptochrome-mediated GA signaling pathway among different species. In response to canopy signals, low blue light-inactivated cryptochrome dissociates from DELLA proteins, and enhances the GA signaling pathway, thus leading to the ubiquitination and GA-induced degradation of DELLA proteins, which further triggers the initiation of leaf senescence. Interestingly, although the direct connection between phytochrome and DELLA proteins remains unclear, the abundance of DELLA proteins reduces drastically under low R:FR [49,50], suggesting that DELLA proteins act as an integrating hub of low R:FR and low blue light signals to modulate leaf senescence (Figure 1).

## 3. Phytohormone-Mediated Leaf Senescence under Shade Conditions

Phytohormones have long been known to affect the onset and progression of leaf senescence, such as ethylene, jasmonic acid (JA), brassinosteroids (BRs), abscisic acid (ABA), and salicylic acid (SA) accelerate leaf senescence. However, auxin and cytokinins (CKs) delay leaf senescence [52]. The content of phytohormones changes drastically in response to the proximity of neighbors, which triggers plant growth adjustments, including enhanced elongation of stems and petioles that enable plants to outgrow their competitors and reach the light (Figure 2) and accelerate leaf senescence to escape from the adverse circumstances [5,8,27].

### 3.1. Ethylene

Ethylene is a vital gaseous phytohormone that plays crucial roles in multiple biological processes, such as seed germination, fruit ripening, and response to biotic and abiotic stresses [53,54]. The regulatory function of ethylene on leaf senescence is well studied. The ethylene biosynthesis genes are upregulated in senescent leaves, accompanied by an increase in ethylene level [55]. Ethylene treatment accelerates leaf and flower senescence, and inhibition of ethylene biosynthesis and signaling could delay senescence [56,57,58]. However, ethylene cannot directly initiate leaf senescence but is a mediator with its downstream ethylene-responsive factors [58,59]. The volatile phytohormone ethylene has long been associated with shade avoidance response. Ethylene not only functions as a neighbor detection signal via atmospheric accumulation but also as a downstream target of photoreceptors in response to vegetation canopy [49,60,61]. Low R:FR treatment induces a consistent upregulation of *ESR2* expression, a marker gene for ethylene signaling that encodes an ethylene receptor protein [62], thus increasing ethylene levels. However, low blue light exposure did not affect ethylene production and the transcription of the ethylene marker gene *ERS2* [49]. Ethylene production reduces in the *pif4* mutant, and PIF5 could directly regulate the transcription of the master transcription factor of ethylene signaling, indicating that the biosynthesis and signaling of ethylene under the regulation of phyB–PIF pathway [30,63]. Furthermore, ethylene delays the GA-induced degradation of DELLA proteins [64], indicating the crosstalk between phytohormone and light signaling pathway to modulate leaf senescence in response to shading. In *Arabidopsis,* the *mir164* mutant displays premature leaf senescence. The expression of microRNA miR164 decreases with increasing leaf age in an ethylene-dependent manner. Through cleavage of *ORE1* mRNA, miR164 represses *ORE1* expression [34], indicating that ethylene is involved in microRNA-mediated leaf senescence.

### 3.2. Abscisic Acid (ABA)

Abscisic acid (ABA) plays a crucial role in seed germination, seedling development, stomatal movements, and leaf senescence, especially under adverse conditions, thus known as the “stress hormone” [65]. Exogenous application of ABA promotes leaf senescence, and ABA content increases in the senescent leaves of rice, maize, and Arabidopsis [66,67]. ABA promotes leaf senescence by enhancing ethylene production. Furthermore, ABA also activates sucrose nonfermenting 1-related protein kinase 2s (*SnRK2s*), which subsequently phosphorylates ABA-responsive element-binding factors (ABFs) and related to ABA-insensitive 3/VP1 (RAV1) transcription factors. The phosphorylated ABFs and RAV1 further upregulate the expression of senescence-associated genes (SAGs) and promote leaf senescence in an ethylene-independent manner [68,69]. ABA content increased under shade conditions in sunflower (Helianthus annuus) and tomato leaves [70,71]. Shade elevates the endogenous ABA level, probably by enhancing the transcription levels of ABA biosynthetic genes, such as nine-cisepoxycarotenoid dioxygenase *3* (*NCED3*) and *NCED5* [72]. Several ABA signaling genes, such as ABF3, are also upregulated in response to neighbor proximity [73].

### 3.3. Brassinosteroids (BRs)

Brassinosteroids regulate various plant growth and development processes, such as photomorphogenesis, seed germination, floral transition, and responses to biotic and abiotic stresses [74]. BR treatment accelerates senescence, and BR-deficient mutants delay the process of leaf senescence [75,76]. Exogenous treatment of epibrassinolide (eBL) altered leaf senescence in a dosage-dependent manner, with low eBL concentrations retarding leaf senescence and high concentrations accelerating this progression of detached wheat leaves [75]. The BR insensitive1 (*bri1*) mutants show a prolonged life span concomitant with a decrease in the expression levels of SAGs. Furthermore, bri1-EMS-suppressor 1(BES1) displays accelerated senescence due to the constitutively active BR response [76]. Overexpressing of UGT73C6, a gene encoding a UDP-glycosyltransferase that inactivates BRs, delays leaf senescence in Arabidopsis [77]. Short-term (4 h) simulated shade conditions lead to lower levels of active BR. However, longer period (24 h) conditions abolish the differences in BR levels, suggesting shade-induced BR levels in a dynamic fashion [4]. Under a vegetational canopy, the low R:FR promotes the nuclear accumulation of phyA. The activated phyA reduced COP1 nuclear speckle, leading to changes in downstream target genes, such as *PIFs* and *HY5*. These targets regulate the BR signaling pathway by altering the expression of BR biosynthesis genes and protein stability of BES1/BZR1 to influence leaf senescence [78]. Interestingly, DELLAs negatively regulate BR signaling by interacting with BZR1 and reducing the transcription of BR-responsive genes [79]. Furthermore, the transcription factor BZR1 physically interacts with each other and synergistically regulates target genes [80].

### 3.4. Strigolactones (SLs)

Strigolactone is a key phytohormone that regulates various physiological processes such as shoot branching, root development, drought tolerance, and leaf senescence [81]. The strigolactone biosynthesis genes *more axially growth 3* (*MAX3*) and *MAX4* were drastically elevated during dark incubation, an extreme shade condition [63]. The transcription level of *branched 1* (*BRC1*), the master repressor of branching, is upregulated under shade conditions [82,83]. BRC1 directly binds to and positively regulates the expression of three HD-ZIP protein-encoding genes, which in turn, together with BRC1, enhance the transcription of ABA biosynthesis gene *NCED3*, leading to the accumulation of ABA and accelerated leaf senescence. Furthermore, under low R:FR conditions, the accumulated PIFs also enhanced the expression of *BRC1* and *NCED3* [84]. Interestingly, exogenous treatment of ethylene and strigolactone simultaneously could markedly accelerate the process of leaf senescence but not by strigolactone only, demonstrating that strigolactone promotes leaf senescence by enhancing the action of ethylene. These results suggest that different phytohormones cowork and interact with each other to regulate leaf senescence in response to shade conditions.

### 3.5. Growth-Promoting Phytohormones Auxin and Gibberellin

Auxin functions in various aspects of cell growth and development [85]. Auxin is known as a negatively acting factor of leaf senescence. Exogenous treatment of auxin inhibits the transcription of SAGs leaf senescence [86]. The expression of IAA biosynthetic genes tryptophan synthase (*TSA1*), IAAld oxidase (*AO1*), and nitrilases (*NIT1-3*) are upregulated in an age-dependent manner. Consequently, the auxin level is elevated during leaf senescence [55]. Studies on the genetic mutation in the auxin signaling pathway further support the involvement of auxin in modulating leaf senescence. Overexpressing of *YUCC6* or gain-of-function mutation *YUCC6*, a gene encoding a flavin-containing monooxygenase that catalyzes the rate-limiting step in the auxin biosynthesis, delay leaf senescence [86]. Disruption of *ARF2*, a transcription repressor in the auxin signaling pathway, causes a delay in leaf senescence. The reduced ARF2 function releases the repression of auxin signaling with enhanced auxin sensitivity, leading to delayed leaf senescence [87].

Gibberellin is well known for its function in regulating cell elongation, seed germination, dormancy, and floral transition [88,89]. The role of gibberellins in leaf senescence has been elusive on account of their involvement in multiple aspects of plant development processes. GA3 (a bioactive form of gibberellins) treatment retards the process of leaf senescence, leading to increased rhizome yield [52]. Exogenous application GA represses leaf senescence, whereas paclobutrazol, an inhibitor of GA biosynthesis, accelerates the process in *paris polyphylla* [90,91]. It is possible that gibberellins are not involved in the regulation of leaf senescence directly but, rather, function by antagonizing the effects of ABA. However, exogenous GA3 treatment was shown to induce the expression of *WRKY45*, a positive regulator of leaf senescence, thus leading to premature leaf senescence [48]. It has been suggested that the role of gibberellins on leaf senescence may rely on the dosage or the situation of the treated leaves [47].

Shade-induced changes in auxin levels have been found in sunflowers and tomatoes, and the transcription of auxin-response genes is dramatically affected after being treated with shade conditions [70,92]. Low R:FR induced the expression of tryptophan aminotransferase of Arabidopsis 1 (*TAA1*), an aminotransferase that catalyzes the formation of indole-3-pyruvic acid from L-tryptophan in the auxin biosynthetic pathway, and TAA1-Related proteins (*TAR*) [93]. On the other hand, low R:FR also stimulates *YUCCA* gene expression through PIF TFs. Thus, shade conditions promote auxin accumulation in the shoot and the elongating hypocotyl [94]. Low R:FR upregulates the expression of *GA20ox1* and *GA20ox2*, which are responsible for the production of bioactive GAs and promotes the accumulation of bioactive GAs [95]. Bioactive GAs interacts with GA receptor GID1, leading to ubiquitination and degradation of DELLA proteins [96]. The abundance of DELLA proteins does decrease in response to the increase in planting density and low R:FR ratio [50].

Confusingly, the increased level of growth-promoting phytohormone is not consistent with the accelerated leaf senescence phenotype under shade conditions. It is possible that shade-induced growth-promoting hormones such as auxin, gibberellin, and cytokinin function in the shoot or emerging apical meristem, promote stem elongation and help a shaded plant to outcompete its neighbors for light-absorbing, not in the mature leaves. However, the senescence-promoting phytohormones, such as ethylene and abscisic acid, function in the mature leaves to accelerate leaf senescence.

### 3.6. Growth-Defense Phytohormones Jamonic Acid and Salicylic Acid

Jasmonic acid (JA) and salicylic acid (SA) are crucial for plant defense against both biotic and abiotic stresses. The exogenous application of methyl jasmonate accelerates the process of leaf senescence by enhancing the transcriptional levels of senescence-associated genes such as *SEN4* and *SAG21* [97]. The transcript abundance of components is responsible for JA synthesis and signaling increases during the initial stage of leaf senescence [67]. Consistent with this, *Arabidopsis* leaves undergoing senescence exhibit 4-fold increase in JA content compared to nonsenescing leaves [98]. Disruption of *3-ketoacyl-CoA thiolase 2* (*KAT2*), the beta-oxidation gene involved in JA biosynthesis, delays dark-induced leaf senescence [99]. JA is thought to integrate stress signals to induce the onset of leaf senescence in an age-dependent manner, given that older leaves experience a more rapid senescence process than younger leaves after exogenous application with methyl jasmonate [52]. Shade treatment can suppress the natural defense mechanisms activated by JA signaling and the expression of genes related to defense [100]. This effect is partially due to the inactivation of phyB by shade, which attenuates JA sensitivity by stabilizing PIFs and JAZs (jasmonate ZIM domains) while destabilizing DELLA proteins. As a result, this facilitates the ability of PIFs and JAZs to activate downstream target genes without being inhibited by DELLA proteins [101,102,103]. Additionally, miR319 can promote leaf senescence by downregulating teosinte branched 1 cycloidea, PCF (TCP) transcription factors, which in turn boosts the level of JA [34].

SA, as a phytohormone known to promote senescence, both initiates and accelerates leaf senescence [104]. Mutant plants, such as the *phytoalexin deficient 4* (*pad 4*) and *nonexpresser of pathogen-related genes* (*npr1*) mutants, exhibit delayed leaf senescence [105]. During senescence, many genes related to SA biosynthesis and signaling are upregulated, and treatment with SA markedly induces the expression of *SAGs*, including *SEN1* and WRKY transcriptional factors *WRKY6*, *WRKY53*, *WRKY54*, and *WRKY70* [52]. When grown under shade, the inactivation of phyB leads to a decrease in SA-mediated defense against pathogen attacks. Low R:FR ratios cause significant changes in the expression of SA-related genes. Plants with impaired SA biosynthesis or signaling are unable to elongate properly in response to Low R:FR ratios [106]. Interestingly, although the content of SA itself remains unchanged under low R:FR light, the degree of SA-dependent phosphorylation of nonexpressor of pathogenesis-related genes 1 (NPR1), a key transcriptional regulator of SA-mediated defense, is reduced [106]. This resource reallocation can attenuate the resistance to abiotic and biotic stresses under shade conditions, as defense mechanisms are hampered by the need for rapid stem elongation.

## 4. Nutrient Deficiency-Mediated Leaf Senescence under Shade Conditions

Light intensity drops drastically for shaded plants [8], and the reduction in photosynthetically-active radiation (PAR) markedly accelerates the process of leaf senescence [7]. Several studies proposed that the induction of leaf senescence under vegetation shading depends more on light intensity than on light quality [107,108,109]. When the light intensity drops below the photosynthetic light compensation point (LCP), it leads to a negative carbon balance which in turn triggers senescence [107]. Low light intensity induces chlorophyll and protein degradation and relocates the nutrients to the developing leaves [108,109]. Thus, low light intensity negatively regulates leaf senescence by influencing the efficiency of carbon fixation and the metabolic balance of carbohydrates (Figure 2).

Light is also crucial for nutrient assimilation and utilization, drop in light intensity and changes in light quality significantly influence this program [110]. Optimal utilization of nutrients accumulated during the photosynthetic period is critical for finely controlling the leaf senescence process. Nutrient limitation under shade conditions was thought as another reason for premature leaf senescence [19]. Nutrient limitation speeds up the hydrolysis and translocation of nutrients from the mature leaves to seeds and emerging organs. The assimilation of nitrate and ammonium was higher when exposure to light than in the dark. Additionally, the expression of several phosphate, sulfate, and K transporter genes markedly increased after plants were shifted from darkness to light conditions, indicating that photosynthetically active radiation is vital for nutrient assimilation [111]. Nitrate reductase (NR) in etiolated buds of peas was similarly induced by white light and red-light illumination, whereas irradiation of far-red or blue light exerted limited extent effects [112], indicating red light plays a dominant role in regulating NR activity.

Light quality also influences the nutrients’ assimilation and utilization in plants. The two light regimes simulated the spectral quality of natural sunlight, and shade light exhibited striking differences in the assimilation and allocation of nitrogen in birch seedlings [113]. Furthermore, short-term shading, dependent on its intensity, results in a rapid decline of phosphorus [114]. Elongated Hypocotyl5 (HY5), a bZIP transcription factor that regulates plant growth and development in response to light, acts as a shoot-to-root mobile signal to help root development and nutrient assimilation [110] (Figure 2). HY5 functions downstream of photoreceptors phytochromes, cryptochromes, and UVR8 to promote photomorphogenesis. Blue light-inhibited leaf senescence is attenuated in the hy5 mutant, indicating that HY5 mediates light-regulated leaf senescence [41]. Photo-excited phytochromes and cryptochromes disrupt and inactive the E3 ubiquitin ligase complexes, which comprise constitutive photomorphogenic 1(COP1) and suppressor of phyA-105 (SPA) proteins, thus allowing the accumulation of HY5 protein. Low R:FR signals indicative of canopy shade are perceived by phytochromes, mainly phyB, leading to the ubiquitination and degradation of HY5 in a 26S proteasome-dependent manner [115,116]. Shoot-derived phyB contributes to red-light regulation of nutrient assimilation by enhancing the accumulation of HY5 in the root. Shoot-derived HY5 activates the transcription of root HY5 and also promotes root nitrate assimilation by inducing *NRT2.1* expression, a gene encoding a high-affinity nitrate transporter [110,117]. ChIP-seq analysis indicated that HY5 could bind to the promoter of a sulfate transporter gene, *SULTR1;2,* via a putative HY5-binding site in the promoter, which was subsequently confirmed by the ChIP assay (Figure 2). In addition, HY5 was found to bind to the promoter of *APR1* and *APR2*, two genes encoding adenosine 5′-phosphosulfate reductase, which are crucial for sulfate assimilation, activating their expression [118]. Phytochrome also enhances the transcription of *APR* genes via the transcriptional regulator phytochrome and flowering time 1 (PFT1), a positive regulator of phytochrome-regulated floral induction in Arabidopsis. Transcription of *APR2* was downregulated in the *pft1* mutant, similar to in the *hy5* mutant. *APR2* expression was more markedly repressed in the *pft1hy5* double mutant [119]. HY5 physically interacts with squamosa promoter-binding protein-like 7 (SPL7), an ortholog of copper response regulator 1 (CRR1) in chlamydomonas, and coworks to activate *miR408*, which promotes the accumulation of plastocyanin, the most abundant Cu-containing protein in chloroplasts, through binding at common G-box sites in the promoter. Cu content in chloroplasts is lower in *hy5*, *spl7*, and *miR408* mutants but higher in the transgenic Arabidopsis plants overexpressing *miR408* compared with the wild type [120]. Thus, HY5 functions as a vital mobile signal, acts downstream of photoreceptors, and mediates external shade signals to affect nutrient assimilation, translocation, and leaf senescence.

## 5. Epigenetic Regulation-Mediated Leaf Senescence under Shade Conditions

Epigenetic processes can change chromatin compaction, influencing the accessibility and binding of transcription factors to cis-regulatory elements in the genome [121]. Chromatin can alter its epigenetic status rapidly and reversely to translate environmental signals such as shade into an acclimation response at the transcriptional level [122]. Exposure of the seedlings to shade, either low light fluence rate or low R:FR ratio, leads to a decrease in chromatin compaction by a reduction in the amount and size of chromocenters [123]. Photoreceptors phyB and CRY2 have been identified as positive regulators of light-dependent chromatin compaction. The *phyB* mutants show lower chromatin density than wild-type plants under normal light conditions [123]. Moreover, phyB and CRY2 control chromatin remodeling complexes responsible for chromatin compaction [124].

Epigenetic modifications such as DNA methylation and histone post-translational modifications play an important role in the transcription of target genes [121]. The increased trimethylation of histone H3 at lysine 4 (H3K4me3) is highly correlated with the induction of *SAGs* during leaf senescence [125,126]. JMJ, a specific H3K4 demethylase, represses leaf senescence through its enzymatic activity. JMJ directly associates with senescence-associated genes *WRKY53* and *SAG201* and suppresses their precocious expression by reducing H3K4me3 levels at these loci in mature leaves [127]. Relative of early flowering (REF6), a histone H3 lysine 27 tri-methylation (H3K27me3) demethylase, promotes leaf senescence by directly binding and activating major senescence regulatory and functional genes such as *nonyellowing 1* (*NYE1*). Consistently, overexpression of SIJMJ4, which specifically demethylates di- and tri-methylations of H3K27, results in premature leaf senescence and promotes dark-induced leaf senescence in tomatoes [128]. However, the direct connection between shade-induced epigenetic modifications and leaf senescence has not been established yet.

## 6. Conclusions and Perspectives

Shade-induced leaf senescence is a highly plastic adaptive strategy when subjected to vegetational shading. Plants speed up their life cycle to escape from the adverse environment, thus ultimately enhancing the possibility of reproductive success [5]. Recent progress has largely extended our knowledge of molecular mechanisms of shade-induced leaf senescence. When shaded by neighboring plants, a drop in light results in a decline in photosynthetic activity and an increase in the expression of *SAGs*. Low R:FR ratio signals are primarily perceived by phytochrome, mainly phyA and phyB. The inactivation of phytochromes by shade enhances the accumulation and activity of phytochrome interacting factors (PIFs), which directly upregulate the expression of the senescence-promoting genes such as NAC transcription factor *ORESARA1*, and increases the levels of senescence-promoting phytohormones such as ethylene, and abscisic acid by modulating the components involved in phytohormone biosynthesis and signaling pathways [30]. The abundance of DELLA protein, a negative regulator of leaf senescence in Arabidopsis, decreases in response to reduced blue light [50]. Cryptochrome 1 (CRY1), the primary blue light receptor, physically interacts with DELLA in a blue light-dependent manner and represses its degradation induced by GA [45]. The levels of senescence-promoting phytohormones, including ethylene, abscisic acid, brassinosteroid, and strigolactone, increased in response to the vegetational canopy. On the one hand, these phytohormones function as a signal of the proximity of neighbors to enable plants to outgrow their competitors and reach the light. On the other hand, they accelerate the decomposition of macromolecules and relocation of nutrients to the emerging tissues or storage organs and achieve reproductive success to escape from adverse light environments. The bZIP transcription factor HY5 acts as a shoot-to-root mobile signal that links the external light environment and photoreceptors signaling to root nutrient assimilation [110]. HY5 is also known to integrate multiple phytohormones, such as abscisic acid and environmental signaling inputs, to fine-tune plant growth and development [129].

Leaf senescence constitutes the final stage of leaf development and tightly links to grain yield, biomass production, and nutritional quality for crop species. Dense planting and intercropping of maize (*Zea mays*) and soybean (*Glycine max*) have been practiced on a large scale in the world and have significant potential to improve crop yield. However, the two planting patterns cause great changes in light environments, such as reduced R:FR ratio and blue light, which trigger severe shade avoidance responses, including exaggerated stem length, reduced stem thickness, lower photosynthetic capacity, and accelerated leaf senescence, and limit the potential for further yield increases. The engineering of photobiology has been shown to be feasible for reducing plant shade avoidance response. For example, enhancing the expression of GmCRY1b, the blue light receptor in soybean, reduces shade avoidance response and exhibits better performance under high-density planting conditions, such as enhanced lodging resistance and markedly increased yield per plant compared with wild-type plants [43]. Shade-induced premature leaf senescence results in decline in net photoassimilates and grain yield. Disruption of *OsNAP*, a plant-specific NAC transcription factor that promotes leaf senescence in rice, results in delayed leaf senescence and a 6.3–10.3% increase in grain yield [130]. Thus, for crop species, extending the functional period of photosynthesis may be an effective strategy to improve yield performance, especially in high-density planting conditions. Studying shade-induced leaf senescence enhances our knowledge of the fundamental biological process and provides means to modulate leaf senescence to improve crop agricultural traits. Engineering the components of photobiology pathways, such as photoreceptors and their interacting partners in the light signaling pathway, can be an essential strategy to obtain crops with ideotypes of leaf senescence to boost yield.

## Figures and Tables

**Figure 1 plants-12-01550-f001:**
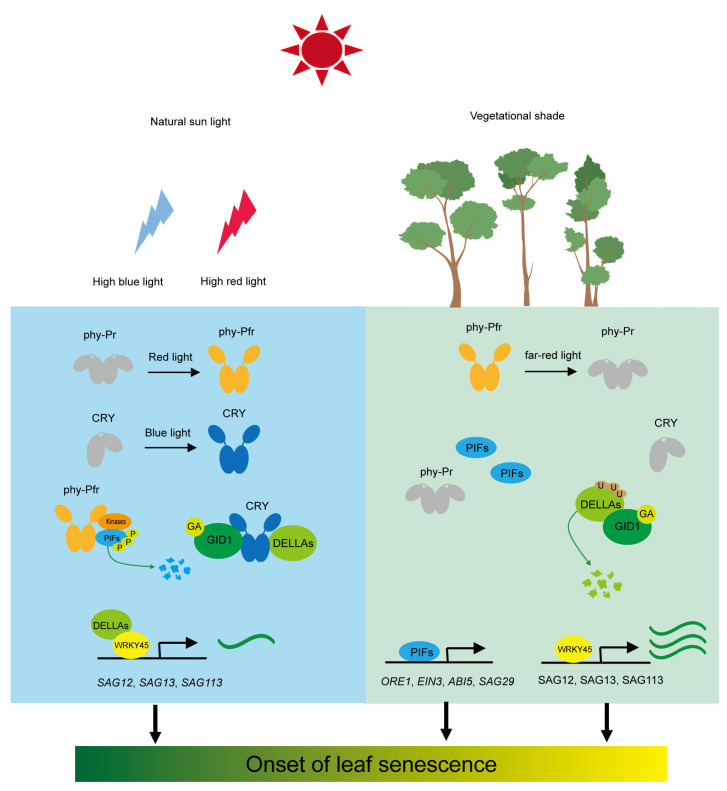
Signaling network of photoreceptor-mediated leaf senescence in response to vegetational shade. Under natural sunlight, high red-light triggers the transformation of the inactive Pr form of phytochromes to the active form Pfr, which then translocates into the nucleus. Light-activated phytochromes interact with transcription factor PIFs, which then undergo phosphorylation and subsequent ubiquitination and degradation through the 26S protease pathway. CRY1 interacts with DELLA proteins in a blue light-dependent manner and enhances their protein stability through repression of GA-induced degradation of DELLA proteins. DELLA proteins physically interact with WRKY45, a positive regulator of leaf senescence, and repress its transcription activation on several senescence-associated genes (SAGs) such as *SAG12*, *SAG13*, and *SAG113*. Under a vegetational canopy, far-red light induces the transformation of the active Pfr form of phytochromes to the inactive form Pr, which further enhances the protein stability of PIFs. PIFs could directly activate the expression of SAGs such as *SAG29*. Furthermore, PIFs accelerate leaf senescence by enhancing the transcription of the components in senescence-promoting phytohormones biosynthesis and signaling. The abundance of DELLA proteins markedly decreases in response to low R:FR ratios and the reduced blue light, leading to the enhanced expression of *WRKY45*. Thin and thick black arrows denote gene expression being repressed and induced, respectively.

**Figure 2 plants-12-01550-f002:**
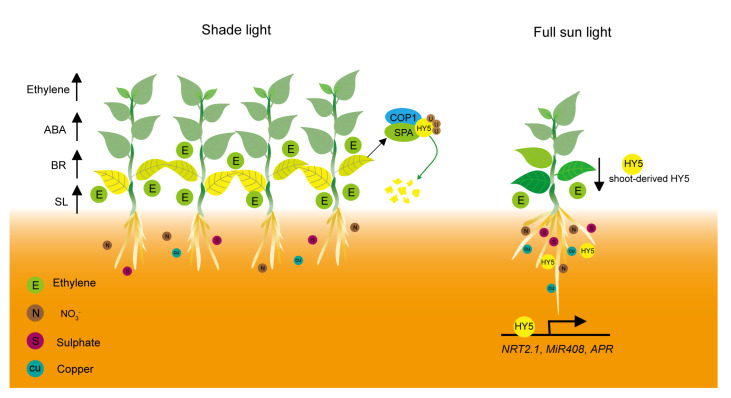
Phytohormones and nutrient deficiency-induced leaf senescence in response to shade. Shade signals increase the levels of senescence-promoting phytohormones such as ethylene, abscisic acid, brassinosteroids, and strigolactones via enhancing the gene expression that is involved in phytohormones biosynthesis and signaling. HY5 acts as a shoot-to-root mobile signal that mediates light promotion of nutrient assimilation. Shoot-derived HY5 activates the expression of *NRT2.1* in root to promote nitrate uptake. Furthermore, HY5 also upregulates the expression of *APR* and *miR408* to promote the assimilation of sulfate and copper. Shading leads to progressive inactivation of phyB and CRYs, resulting in enhanced activity of the E3 ligase complex COP1/SPA, which controls the ubiquitination and subsequent degradation of transcription factor HY5. Upward arrows denote the increased levels of phytohormones, and the downward arrow indicates shoot-derived HY5 translocate into the root.

## Data Availability

No new data were created or analyzed in this study. Data sharing is not applicable to this article.

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
