# Peer review of "Shade-Induced Leaf Senescence in Plants"

_plants, 2023, doi:10.3390/plants12071550_

Round 1

Reviewer 1 Report

It is written very clearly and logically. The current state of knowledge is presented.

Perhaps, in my opinion, there is a lack of even a brief discussion of the role of polyamines in dark-induced senescence of leaves. I think it would be worth supplementing the work with this aspect.

Other than that, I have no comments or remarks.

Author Response

Point 1: There is a lack of even a brief discussion of the role of polyamines in dark-induced senescence of leaves. I think it would be worth supplementing the work with this aspect.

Response: Thank you for dedicating your precious time in reviewing our manuscript and providing constructive feedback. Based on your suggestion, we have incorporated the role of polyamines in dark-induced senescence of leaves in the appropriate section (page 3, lines104-108).

Reviewer 2 Report

The food crisis is sweeping the world and the number of people affected by hunger is increasing globally. The increase in crop yield is not keeping up with the population growth. The paper by Li et al. reviews the mechanisms of leaf senescence under shade conditions from three aspects: photoreceptors, hormones and nutrients. The MS provides effective strategies and design concepts for highly dense crops and for extending the duration of photosynthesis in leaves. It provides theoretical basis and strategic guidance for improving crop yield.

This MS is very meaningful and it is recommended to be accepted.

Manor issues:

1. It is recommended to supplement the pathways of leaf senescence regulated by other hormones such as salicylic acid and jasmonic acid in the third part.

2. Suggesting to complement the regulation of leaf senescence other than genetic coding. Such as epigenetic regulation, small RNA mediated leaf senescence, and cite the more publications.

Minor issues:

3. Check line 107 on page 3 that the light-harvesting chlorophyll a/b-protein complex II (LHC II) should be light-harvesting complex II (LHC II);

4. Check Figure 1 and Figure 2. The gene name in the figure is not in italic format;

5. Check page 6, line 210, 2.1 should be 3.1;

6. Check the abbreviation of microRNA408 and unify the whole text;

7. Check the full text reference format and journal name to ensure they are consistent. For example, references 9, 25, 28, 31 and 111 have different formats of journal names.

Author Response

Manor issues:

1. It is recommended to supplement the pathways of leaf senescence regulated by other hormones such as salicylic acid and jasmonic acid in the third part.

 Response: Thanks for your suggestion. We have supplemented relevant contents about phytohormones salicylic acid and jasmonic acid in regulating leaf senescence in appropriate sections (Page 8, lines 341 to 375).

2. Suggesting to complement the regulation of leaf senescence other than genetic coding. Such as epigenetic regulation, small RNA mediated leaf senescence, and cite the more publications.

Response: Thanks for your suggestion. We have added relevant contents about epigenetic regulation in the revised manuscript (page 10, lines 448 to 472). The contents of small RNA-mediated leaf senescence were added in appropriate sections (page 6, lines 236 to 240, and page 8, lines 358 to 360).

Minor issues:

3. Check line 107 on page 3 that the light-harvesting chlorophyll a/b-protein complex II (LHC II) should be light-harvesting complex II (LHC II);

Response: We have substituted the description of “light-harvesting chlorophyll a/b-protein complex II (LHC II)” with “light-harvesting complex II (LHC II)”. Page 3, line 113.

4. Check Figure 1 and Figure 2. The gene name in the figure is not in italic format;

Response: The font of all the gene names in Figure 1 and Figure 2 have been replaced with italic format.

5. Check page 6, line 210, 2.1 should be 3.1;

Response: Thanks for your reminder. We have already replaced 2.1 with 3.1. Page 6, line 216.

6. Check the abbreviation of microRNA408 and unify the whole text;

Response: We have unified the abbreviation of microRNA in the whole text, such as the abbreviation of microRNA408 is replaced with miR408.

7. Check the full text reference format and journal name to ensure they are consistent. For example, references 9, 25, 28, 31 and 111 have different formats of journal names.

Response: We greatly appreciate for reminding us about the consistency of the reference and we have carefully checked and corrected relevant mistakes.

Reviewer 3 Report

This review give the great information about shade condition and leaf senescence. I have minor recommendation as following;

Abstract: Current abstract did not give enough detail about shading and senescence. It like only introduction. Authors should rewrite the abstract.

P.2, first paragraph of 2.Photoreceptor...: It is better if authors give some details about photoreceptors and their example such as phytochrome before descript it in 2.1 and 2.2.

Fig1 and Fig2: They should refer in the context. 

Author Response

Point 1. Abstract: Current abstract did not give enough detail about shading and senescence. It like only introduction. Authors should rewrite the abstract.

Response: Thanks for your suggestion. We have rewritten the abstract and added more information about shading and leaf senescence.

Point2. P.2, first paragraph of 2.Photoreceptor...: It is better if authors give some details about photoreceptors and their example such as phytochrome before descript it in 2.1 and 2.2.

Response: We have added relevant descriptions about the indicated photoreceptors.

Point3. Fig1 and Fig2: They should refer in the context. 

Response: Fig1 and Fig2 have been cited in the appreciate section of this context.